# Characterizing Touch Discrimination Impairment from Pooled Stroke Samples Using the Tactile Discrimination Test: Updated Criteria for Interpretation and Brief Test Version for Use in Clinical Practice Settings

**DOI:** 10.3390/brainsci13040533

**Published:** 2023-03-23

**Authors:** Yvonne Y. K. Mak-Yuen, Thomas A. Matyas, Leeanne M. Carey

**Affiliations:** 1Occupational Therapy, School of Allied Health, Human Services and Sport, La Trobe University, Melbourne 3086, Australia; y.mak-yuen@latrobe.edu.au (Y.Y.K.M.-Y.); t.matyas@latrobe.edu.au (T.A.M.); 2Neurorehabilitation and Recovery, Florey Institute of Neuroscience and Mental Health, University of Melbourne, Melbourne 3084, Australia; 3Department of Occupational Therapy, St Vincent’s Hospital Melbourne, Melbourne 3065, Australia

**Keywords:** cerebrovascular accident, somatosensory, touch, standardized assessment, upper extremity, hand

## Abstract

Somatosensory loss post-stroke is common, with touch sensation characteristically impaired. Yet, quantitative, standardized measures of touch discrimination available for clinical use are currently limited. We aimed to characterize touch impairment and re-establish the criterion of abnormality of the Tactile Discrimination Test (TDT) using pooled data and to determine the sensitivity and specificity of briefer test versions. Baseline data from stroke survivors (*n* = 207) and older neurologically healthy controls (*n* = 100) assessed on the TDT was extracted. Scores were re-analyzed to determine an updated criterion of impairment and the ability of brief test versions to detect impairment. Updated scoring using an area score was used to calculate the TDT percent maximum area (PMA) score. Touch impairment was common for the contralesional hand (83%) but also present in the ipsilesional hand (42%). The criterion of abnormality was established as 73.1 PMA across older adults and genders. High sensitivity and specificity were found for briefer versions of the TDT (25 vs. 50 trials; 12 or 15 vs. 25 trials), with sensitivity ranging between 91.8 and 96.4% and specificity between 72.5 and 95.0%. Conclusion: Updated criterion of abnormality and the high sensitivity and specificity of brief test versions support the use of the TDT in clinical practice settings.

## 1. Introduction

Stroke affects an estimated 17 million people each year globally and is the second largest cause of death and disability worldwide [1]. Loss of body sensations is experienced in one out of every two survivors of stroke [2], with negative impacts on survivors’ ability to interact with the world around them [3,4], goal-directed arm use [5,6], and return to previous life activities [7,8,9]. Quantitative, standardized, and precise measurements with good discriminative validity are necessary to diagnose impairment, establish criteria that can be used to evaluate the effectiveness of treatment, and monitor the progress of change [7,10]. However, many therapists are using non-standardized measures in the clinical setting [11,12]. Clinical measures commonly used often have poor reliability and sensitivity and have a reduced ability to identify the presence of impairment compared to quantitative standardized tests [13,14].

Improved detection of somatosensory impairment in the rehabilitation setting is important given the impact of these impairments on daily activities [7] and the need to inform clinical pathways and treatment services [15,16,17]. For a test to be useful in clinical practice, it needs to target clinically important outcomes, be sensitive enough to detect change, be acceptable to the end users (e.g., rehabilitation physicians, therapists, and stroke survivors) and be practical for use in clinical settings. Assessment of the ability to discriminate differences in textured surfaces is important after stroke [2,18,19].

Valid, quantitative measurement is critical to diagnose somatosensory impairment and assess change over time [18]. Some measures specifically assess for touch discrimination [20,21,22]. One quantitative, standardized measure that is available to assess touch discrimination function post-stroke is the Tactile Discrimination Test (TDT) [23]. The TDT has a pedigree of development in primate studies [24] and in neuroimaging studies with humans [17,25]. Using the TDT, we have demonstrated evidence of involvement of the somatosensory system in stroke survivors with somatosensory impairment [26,27], evidence of clinical improvement in touch discrimination with training [18,26,28], and evidence of change in somatosensory brain regions and networks with impairment and recovery [13,29,30], providing support for the construct validity of the measure. The test has been recommended within the National Institute of Health Toolbox project [31]. 

The TDT is suitable for use in research and clinical practice settings [18,23,26,28,31]. The TDT requires participants to discriminate differences in finely graded plastic texture grid surfaces using a three-alternative forced-choice design and has high retest reliability, age-appropriate normative standards, and good discriminative properties [23]. However, a reduction of test time is recommended for clinical practice settings if high sensitivity and specificity are retained. The original test version consisted of 50 trials (comprising 10 trials of each of the five stimulus sets) [23]. Subsequent clinical trials used 25 trials (5 trials of each stimulus set) within their research protocols [26,27,29,30,31,32]. A reduced number of trials was used to reduce the testing time (approx. 10 min rather than 20 min for the 50 trials) and minimize fatigue. In the current study, we therefore sought to investigate the sensitivity and specificity of a reduction in the number of trials administered: specifically, 25 vs. 50 trials; 15 vs. 25 trials; and 12 vs. 25 trials. The 12-trial version (three trials of four stimulus sets) was explored based on the observation that the finest texture set of 1550 µm (micrometers) was correctly discriminated from the 1500 µm anchor stimulus only with chance probability (0.33) based on normative data sets. Hence a reduction in the number of stimulus sets was also explored based on deleting the 1550 µm triplet set from the test, resulting in 12 trials.

In recent years, progress has been made in updating scoring for the TDT. The original version of the TDT required the presentation of five triplet surface sets. The probability of correct response for each level of stimulus difference was calculated, and a discrimination limen, known as the percent spatial increase (PSI) score [23], was determined. However, it was observed that responses did not always follow the expected ogive (sigmoid) function in survivors of stroke, especially when the number of trials was reduced [29]. The use of an area score that uses all available response data was investigated. The area under the psychometric function has advantages as a quantitative index to measure overall TDT response. Such an index does not assume the function reaches any particular asymptote and does not become indeterminate when the data does not permit the location of the limen due to lower upper asymptotes or data sets that do not permit the location of a limen for any reason [33]. Quantification of TDT scores by computing an area under the psychometric function is also likely to be more achievable in clinical practice settings. With the requirement of a valid and practical test for use in clinical practice settings, the TDT scoring was updated, with the TDT score now referred to as the TDT percent maximum area (PMA) score [33]. 

In the present study, we sought to undertake a detailed pooling of baseline TDT data from six completed studies (six with survivors of stroke; and four with neurologically healthy adult controls) to:characterize the distribution, frequency, and severity of impaired touch discrimination, based on the quantitative TDT, in a larger stroke sample;establish updated normative standards and criteria of abnormality using the new PMA scoring;compare the distribution and frequency of impaired stroke TDT scores across test versions (50, 25, 15, and 12 trials);determine the sensitivity and specificity for briefer versions of the TDT (25, 15, and 12 trials) to accurately assess the presence of somatosensory loss in stroke survivors.

## 2. Materials and Methods

### 2.1. Study Design and Samples

This study involved pooling raw paper-based or electronic files of baseline TDT data from six independent studies conducted by Carey and colleagues. Baseline data were pooled for stroke survivors (*n* = 207) and for neurologically healthy people (*n* = 100). Stroke survivor data were pooled from the Discriminative Validity study [23], Study of the Effectiveness of Neurorehabilitation on Sensation (SENSe) trial [26], the Imaging Neuroplasticity of Touch (IN_Touch) study [29,30], the Connecting New Networks for Everyday Contact Through Touch (CoNNECT) study [27,34], additional testing linked with the National Institute of Health (NIH) Toolbox study [31], and the SENSe CONNECT Study [32].

Stroke participants had a diagnosis of first stroke and included participants with and without somatosensory loss, according to standardized clinical assessments. The Discriminative Validity and NIH Toolbox studies were measurement-focused and therefore recruited participants with and without sensory loss [23,31]. The remaining studies included sensory rehabilitation, and so participants were screened for somatosensory impairment in one or more modalities before being recruited. Testing across modalities was conducted at baseline before any intervention [26,27,29,30,31,32,34]. All participants could understand 2–3 step instructions and provided informed consent to participate. Stroke survivors were excluded if they were medically unwell, had central nervous system dysfunction other than stroke, diagnosis of peripheral neuropathy, or presence of unilateral spatial neglect. Participants in the IN_Touch and CoNNECT trials also needed to be right-hand dominant, first-ever stroke patients who were able to undergo Magnetic Resonance Imagining (MRI) [27,29,30,34]. Participants were at least 6 weeks post-stroke for the SENSe trial and 12 weeks post-stroke for the CoNNECT trial. Participants in the SENSe CONNECT trial who had more than one stroke were excluded from the pooling of data to ensure consistency.

Data from neurologically healthy adult controls (*n* = 100) was pooled from the Normative Validity study [23], the IN_Touch study [29,30], the CoNNECT study [27,34], and from additional testing linked with the NIH Toolbox study [31]. Healthy participants without any history of neurological dysfunction or impairment of sensibility in the upper limbs were recruited. There was no overlap of healthy participants across studies.

Raw data were pooled across studies as studies recruited participants with similar stroke or healthy characteristics and utilized the same TDT and standardized protocol to determine the level of tactile discrimination ability. Research therapists were trained in the administration of the TDT assessment and blinded to intervention conditions. Demographic information, such as age, gender, and hand dominance, was extracted from each participant’s research file. The current project was approved by Austin Health HREC/17/Austin/281 and La Trobe University Human Ethics Committees, Melbourne, Victoria, Australia.

### 2.2. Measure: Tactile Discrimination Test (TDT)

The TDT is a quantitative measure of the ability to discriminate differences in finely graded texture surfaces using a three-alternative force choice design [23] (see Figure 1). With vision occluded, participants were presented with three texture grids (i.e., a triplet set) in the window opening of the device. Participants either feely explored the texture surfaces with their nominated preferred fingertip (usually of index or middle finger) or were guided over the surfaces using a standardized procedure if movement was limited. Participants were required to determine the odd surface out, i.e., the texture grid that differs, that could be in the first, second, or third position. The original full version required the presentation of 50 trials of the triplet sets [23]. The current test version typically required the presentation of 25 trials of surface triplets [29]. The final test score is the percent maximum area (PMA) which accounts for the probability of correct discrimination response across all stimuli presented and represents the area that subtends the psychometric function after accounting for chance [33].

### 2.3. Data Analysis

Quantitative analyses of pooled data were conducted using Microsoft Office Excel 2016 and IBM SPSS Statistics (Version 28.0; IBM Corp., Armonk, NY, USA). Descriptive statistics were used to characterize TDT scores and included the mean/median, frequency distributions, range/spread, and standard deviation of scores. The distribution of scores from the contralesional and ipsilesional hands of stroke participants and the dominant and non-dominant hands of healthy participants were examined with histograms. The presence of impairment was identified relative to the criteria of abnormality defined using the neurologically healthy sample.

The criterion of abnormality was defined as the fifth percentile of the neurologically healthy sample for the TDT (higher scores represent better performance). A more conservative criterion in which all TDT scores of the sample were contained was also defined as the zero percentile. These criteria were determined separately for dominant and non-dominant hands and for the combined hands of healthy controls. They were determined and confirmed with reference to stroke distributions and were then used as a reference point for interpreting TDT scores [37]. A zone of uncertainty around the criterion of abnormality [23] was also calculated using the standard error of measurement to guide the interpretation of impairment. Quantile regression was used to verify the impact of age on the location of the fifth percentile.

The association between scores from the briefer test versions compared to the original full or 25-trial test versions for the contralesional hands of stroke subjects was examined using scatterplots and used to determine the linear relationship between the briefer and original or 25-trial test versions [37]. Contingency table analyses were subsequently conducted using these data to analyze the sensitivity and specificity for adopting briefer versions of the TDT compared to the TDT25 trial test version [37].

## 3. Results

### 3.1. Background Characteristics of the Samples

Data files of 207 neurologically impaired stroke participants from six studies were involved in this analysis. Ages ranged from 18 to 87 years. The mean age was 56.4 years, with a standard deviation of 14.5 years, and 178 participants were right-hand dominant. The demographic characteristics of the pooled stroke sample involved in this analysis are shown in Table 1.

Data files of 100 neurologically healthy controls pooled from four studies were involved in this analysis. Participant characteristics were similar between trials, as shown in Table 2. The ages of participants ranged from 23 to 89 years. The mean age was 52.7 years, with a standard deviation of 16.2 years. Ninety-six participants were right-hand dominant.

### 3.2. Frequency Distribution of Tactile Discrimination in Pooled Sample of Healthy Older Adults (n = 100)

Frequency distributions of TDT scores for neurologically healthy participants are presented in Figure 2. The mean score for the dominant hand was 86.45 PMA (6.89 PMA, SD) and 84.71 PMA for the non-dominant hand (6.85 PMA, SD) for the TDT25.

### 3.3. Updated Normative Standards and Criteria of Abnormality

Of the four studies, only two (Normative Discriminative study and SENSe) used the full original test versions (i.e., ten repetitions of the five stimulus sets; *n* = 50 trials; *n* = 63 combined samples for Normative Discriminative study). The other two studies (CoNNECT and NIH) used the briefer, 25-trial versions only (i.e., five repetitions of each of the five stimulus sets). Correlation analysis of ten vs. five repetitions for the combined sample (*n* = 63) yielded a correlation of *r* = 0.74 when dominant and non-dominant hand data were combined (Figure 3).

To ensure a maximum sample size within the pooled data analysis, data will be presented for the 25-trial version (five repetitions of the five stimulus sets; TDT25). In addition, briefer trial versions will be investigated, namely the 15-trial version (three repetitions; TDT15) and 12-trial version (three repetitions of four stimulus sets, not including the finest grid comparison; TDT12).

Based on the 25 trials (*n* = 100), the TDT scores for the combined dominant and non-dominant hand data of the healthy sample (Table 3) had a mean PMA of 85.58 and a standard deviation of 6.91 PMA. The mean PMA of combined hands is similar for the TDT15 trial test version, but it is higher than the TDT12-trial version, in which only four stimulus sets were presented. As expected, variation in TDT scores was greater with fewer trials. The fifth percent criterion of abnormality for TDT25 was 74.47 PMA for the dominant hand, 72.76 PMA for the non-dominant hand, and 73.10 PMA for the combined data. The zeroth percentile criterion for abnormality was defined as 63.45 PMA.

The criterion of abnormality range for the TDT25 was conservatively estimated based on a re-estimation of reliability and standard deviation of the observed values for the pooled stroke data. Reliability values for calculation of the standard error of measurement were re-estimated for the TDT25 using the Spearman–Brown Prophecy Formula [38,39]. The estimate of reliability for the shorter test version was 0.85 (rather than 0.92 reported for the TDT50 [23]). The SEM was conservatively estimated using the SD of the TDT25 observed scores, i.e., 20.57 PMA (Table 4). Based on this data, the SEM was 7.97 PMA. Given a one-tailed estimate for the presence of impairment, the confidence interval for the zone of uncertainty for the fifth percentile criterion of abnormality for the TDT25 was estimated to be 73.10 +/− 13.14 PMA for an individual score.

Factors that could potentially influence TDT scores of neurologically healthy participants, such as hand dominance and age, did not demonstrate statistically significant effects. Within the limits of statistical error, there was no support for an age effect on either the dominant or non-dominant side, and the models for the two sides showed slope coefficients that could not be differentiated within statistical limits. The resulting model for the 0.05 quantile on the pooled sides data indicated virtually zero slope, thus failing to support a case for an age-adjusted impairment threshold. The same methodology applied to the TDT25 followed a similar story: 73.1 PMA was again obtained as an estimate after pooling sides. No gender effect was discerned within the limits of statistical error when estimates were obtained separately for the dominant and non-dominant sides. Confidence intervals for the fifth percentile strongly or completely overlapped for males and females on both dominant and non-dominant hands. Considering this evidence, the data were pooled over the two hands to explore if increasing the sample sizes revealed, via more precise confidence intervals, a gender effect. Confidence intervals nevertheless remained largely or completely overlapped between males and females for all variants of the test.

### 3.4. Frequency Distributions of Tactile Discrimination in Pooled Sample of Stroke Survivors (n = 207)

The frequency distributions of TDT25 scores for stroke participants are presented in Figure 4. The mean score for the contralesional hand of the pooled data (*n* = 207) was 50.50 PMA (20.57 PMA, SD) and for the ipsilesional hand was 74.14 PMA (12.43 PMA, SD). Based on the fifth percentile criterion of abnormality for the TDT25, i.e., 73.10 PMA, 84% of the total sample (174/207) was defined as having impairment in the contralesional hand. Scores ranged from 8.28 PMA, indicative of most severe impairment, to 98.97 PMA, well within the ‘healthy’ performance range. Impaired performance of the ipsilesional hand was also present in 47% (98/207) of the sample. Scores ranged from 35.86 PMA to 97.93 PMA.

Descriptive statistics for TDT scores of the stroke sample for the contralesional and ipsilesional hands for original and brief versions of the TDT are shown in Table 4. A median total score of 49.31 PMA on the TDT25 (contralesional side) is considerably lower than the median total score achieved on the ipsilesional side (74.14 PMA) and in healthy controls (85.86 PMA). Median scores ranged from 48.62 to 53.85 across trial versions for the contralesional hand and from 74.14 to 78.21 for the ipsilesional hand.

### 3.5. Association between Test Scores for Brief Test Versions Relative to Original TDT50 and TDT25 Test Versions

Scatterplots were used to display the spread and nature of the relationship between scores for briefer test versions and the original TDT50-trial version (*n* = 118) for the contralesional hand of survivors of stroke. As not all participants completed the TDT50, the analysis was primarily completed for the briefer test versions and the currently used TDT25-trial version (*n* = 207). There was a positive association between all brief versions and the TDT25. The strength of the association was high for all test versions (*r* = 0.92 to 0.94), with briefer trial versions showing slightly lower correlations as expected (Figure 5).

### 3.6. Sensitivity and Specificity for Briefer Versions of the TDT (25, 15, and 12 Trials)

The presence and nature of agreement and disagreement between test scores are indicated in the contingency table analysis (Figure 6). This analysis presents predictions of impairment when the fifth percentile criterion of abnormality is employed for the TDT.

The TDT25, when compared to the TDT50, had a sensitivity of 91.8% and specificity of 95%, based on scores for the contralesional hand and the fifth percentile criterion of abnormality. In comparison, the briefer TDT15, when compared to TDT25, had a sensitivity of 93.5% and a specificity of 81.6%. Further analysis found a high sensitivity in the 12 vs. 25 trials (96.4%), with a specificity of 72.50%.

## 4. Discussion

Using a pooled data sample, our findings clearly demonstrate the frequency, and severity of touch discrimination impairment via the ability of the TDT to identify impairment in touch discrimination in survivors of stroke relative to age-matched neurologically healthy adults. The affected contralesional hands of stroke survivors were frequently impaired, with impairment scores ranging from just impaired (relative to healthy controls) to severely impaired (maximum impairment score, equivalent to or worse than chance discrimination). In addition, the contralesional hand was more frequently and severely impaired than the ipsilesional ‘unaffected’ hand, as expected from the typical stroke syndrome [2,40,41]. The presence of impairment in the ipsilesional hand is consistent with findings that somatosensory disturbances may occur bilaterally [14] and with our earlier findings [2]. While the ipsilesional hand is often assessed in clinical settings as a point of reference for unimpaired performance, our findings highlight that this assumption is often not true.

Our analysis of pooled datasets of healthy controls permitted the definition of updated criteria of abnormality for all test versions based on a larger sample size. In turn, this allowed for identification of the presence of tactile impairment with greater accuracy and confidence. Objectively defined normative guidelines are important to be used as a point of reference within clinical practice settings. This is particularly the case given evidence of potential impairment in the ipsilesional ‘unaffected’ hand and current clinical practice where the ipsilesional limb is often used as a point of reference for unimpaired TDT scores.

Age was not significantly associated with touch discrimination score in the older healthy samples. This finding suggested that it was appropriate to combine the normative data across ages (i.e., 23 to 89 years) when interpreting TDT scores. Similarly, when dominant and non-dominant hands were compared in healthy individuals across all ages and genders (male and female), we did not find significant differences. Thus, there is currently no contrary argument around a fixed criterion percentile. While the ipsilesional hand is often assessed in a clinical setting as a point of reference for ‘unimpaired’ performance, evidence of impairment in a relatively large proportion of stroke survivors (42%) suggests this is problematic. Given time limitations, the briefer version of the TDT may be used clinically, given the TDT has a defined criterion of abnormality to guide interpretation.

Comparing the TDT50 and TDT25 trial versions showed a high correlation *r* = 0.94, with 88.6% of the variance when only using the first half of the test, suggesting the TDT25 is a good substitute for the TDT50 trial test version within clinical settings as the latter is more resource intensive. The criterion of abnormality for TDT25 (based on the fifth percentile) was found at 73.1 PMA. The pooled average criterion of abnormality is 73.1 PMA (dominant hand = 74.47 PMA; non-dominant hand = 72.76 PMA) for older neurologically healthy adults.

The sensitivity and specificity findings appear promising. Best estimates from the pooled samples suggest the ability for the TDT to be implemented using briefer test versions with high sensitivity and specificity. A strong positive association has been demonstrated (*r* = 0.92–0.94) between the TDT25 and TDT50 versions of the TDT, with only 8/118 missed impairments and 1/118 inaccurately diagnosed. When compared to the clinical TDT25, the number of participants with missed impairments using the TDT15 was low, with a miss rate of 3%. A further reduction in the number of trials was achieved by deleting the stimulus set with the finest texture grating of 1550 μm, as discrimination of this fine difference was found to typically occur by chance [23]. The number of disagreements made using the TDT12 was low, with 5.3% of individuals incorrectly identified as unimpaired. Inspection of the missed impairments indicated that almost all had scores within the area of uncertainty around the criterion of impairment and thus may represent measurement error or only relatively mild impairment. With relatively high sensitivity (96.40%) and moderate specificity (72.50%) compared to other briefer trial versions, the presentation of TDT12 appears capable of identifying (or screening for) impairments accurately in most stroke patients, particularly in those who have moderate to severe sensory impairment post-stroke. Thus, both TDT15 and TDT12 trial versions meet the criteria of good sensitivity and brief presentation time (based on the brief number of trials) and have the potential for use in the clinical setting.

A previous study by Carey et al. (2002) using the 50-trial version and best available clinical measures of texture discrimination revealed that impairment was missed in 25% of patients when the clinical measure was used (based on the fifth percentile criteria of abnormality for the TDT). In addition, the clinical test incorrectly identified impairment in 40% of patients [13]. Most of the patients included in the Carey et al. 2002 study were also included in the current study. This provides an opportunity to indirectly compare false alarm and miss rates of the clinical measures with those using the brief test versions in the current study. The clinical test of texture discrimination was based on the correct matching of 10 common textures. In comparison, the overall miss rate for the TDT12 was much lower (5.3%), and there were very few inaccuracies providing evidence of the enhanced ability of the TDT12 to detect impairment. Inadequate detection of impairment based on clinical measures was also highlighted in the study conducted by Kim and Choi-Kwon (1996) in acute stroke patients, with approximately half of the patients identified as impaired being missed with routine clinical measures [14]. This again reinforces the need for a quantitative and sensitive measure of texture discrimination, such as the tactile discrimination measure used by Kim and Choi-Kwon, which was a modification of the original TDT test [23].

The spread of total PMA scores across the full range of possible scores, from unimpaired TDT scores to most severe tactile impairment across the TDT25, TDT15, and TDT12, suggests that the 15- and 12-trial versions would perform well in a clinical practice setting. The wide spread of scores, shown in Figure 2, is not unexpected given the high variability in stroke severity and lesion location. This spread suggests the presence of a range in severity of impairment, consistent with existing literature [2,10,16,18,42]. An increase in frequency at scores greater than 73.10 PMA for 25 trials and 69.45 PMA for 15 trials is indicative of unimpaired TDT scores relative to healthy controls.

Findings from our current pooled sample appear to be generalizable to the wider population of stroke survivors, at least those in the subacute to chronic phases of recovery. Our pooled data was relatively heterogenous, including survivors of stroke with cortical and/or subcortical lesions, right or left hemisphere lesions, and ischemic or hemorrhagic stroke. There were more men than women, and the mean age was lower than the general population of stroke survivors [43], although the burden of stroke in people younger than 65 years has increased over the last few decades [1]. The pooled sample included people who had either their dominant (49%) or non-dominant (51%) hand affected and were at varying times post-stroke, ranging from 2 to 990 weeks post-stroke. As such, they may be considered comparable to the population of stroke survivors who present for rehabilitation [2] and/or may be living in the community.

Findings from the current study provide important information for occupational therapists and physiotherapists working in stroke rehabilitation. This study highlights the ability to shorten the current quantitative TDT test to user-friendly, time-efficient versions, thus assisting in assessing for discriminative touch sensation in the stroke population. The findings provide evidence that briefer versions of standardized, quantitative tests are capable of efficiently testing for somatosensory loss, at least in the touch modality and using the TDT, thus advancing current measures of tactile discrimination in a clinical setting. Furthermore, despite TDT trials being delivered by a range of assessors (originator of the test as well as less experienced researchers/clinicians), sensitivity and specificity have remained high.

The TDT advances current clinical measurement of tactile discrimination as it has quantitative scales, norm-referenced standardized test scores, and can discriminate the presence of impairment. It is recommended that the TDT25 be used as the ‘new gold standard’ with briefer test versions providing clinicians with a quantitative test that is clinically applicable to post-acute, rehabilitation stroke patients. Furthermore, the age-matched TDT score standards established for the test can be used as a frame of reference from which clinicians can interpret stroke survivors’ test scores relative to older neurologically healthy individuals. In addition, the brief test versions for the TDT can provide clinicians with quantitative information to guide subsequent somatosensory retraining, directly influencing the rehabilitation process.

### Limitations

The estimate of the zone of uncertainty for the new criterion of abnormality was conservatively estimated based on existing data. It is recommended that the reliability index be recalculated based on observed reliability data obtained using the new PMA scoring. Findings from the current study may be generalized to stroke participants in the rehabilitation setting but not necessarily in acute settings. Raw data collected from participations who were involved in previous stroke studies were from post-acute, rehabilitation and chronic stroke patients. Participants were not based in an acute setting. In addition, the generalization of results to stroke populations in other geographical locations may be limited by the sample investigated. The sample was primarily derived from survivors of stroke in metropolitan Melbourne and limited to diversity in cultural groups (all participants were English-speaking). This may not be an accurate representation of the full Australian or other geographically diverse stroke populations. Although a more culturally diverse sample would have been ideal, the TDT was administered in English, and funds were not available for interpreters. Finally, the sample was also selected on the basis of the selection criteria. For example, they required a minimum level of comprehension, and individuals with neglect were excluded. This, again, could potentially influence the generalization of findings to other stroke populations.

## 5. Conclusions

Survivors of stroke may present with touch discrimination impairment in the contralesional hand, as expected, but also in the ipsilesional hand. The presence of touch discrimination impairment is common after stroke (83% in this pooled sample), with impairment ranging from just noticeable impairment relative to age-matched healthy controls to severe impairment, with discrimination of texture differences less than chance. Further, impairment in the ipsilesional hand is relatively frequent (42% in this sample), especially in those who experienced contralesional impairment.

The TDT has the potential to accurately and efficiently identify tactile discrimination impairment in clinical practice settings. The updated criterion of abnormality of the TDT may now be used to guide interpretation in research and clinical practice settings. In addition, brief versions of the quantitative Tactile Discrimination Test (TDT25, TDT15, and TDT12) may be used to identify impairment with high sensitivity and specificity in survivors of stroke. With strong empirical and psychometrical foundations, as well as time efficiency in administration, this measure should supplement or replace existing clinical measures that have been shown to be inadequate in identifying the presence of impairment.

## Figures and Tables

**Figure 1 brainsci-13-00533-f001:**
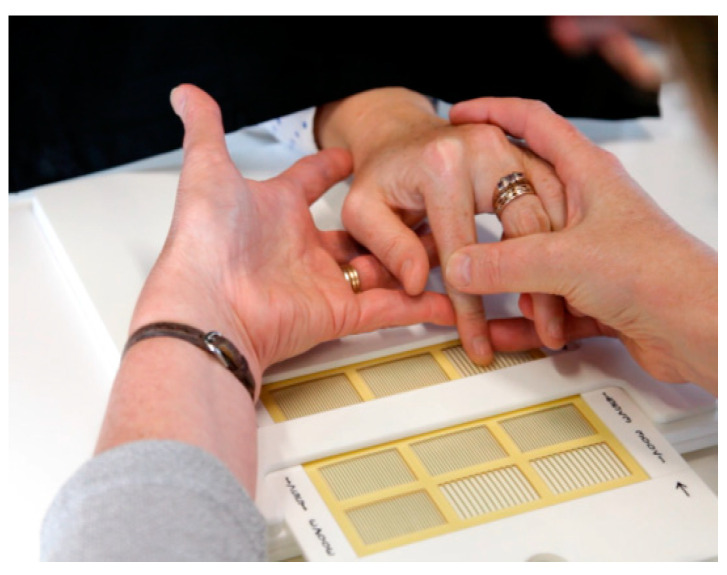
Tactile Discrimination Test (TDT) [35,36].

**Figure 2 brainsci-13-00533-f002:**
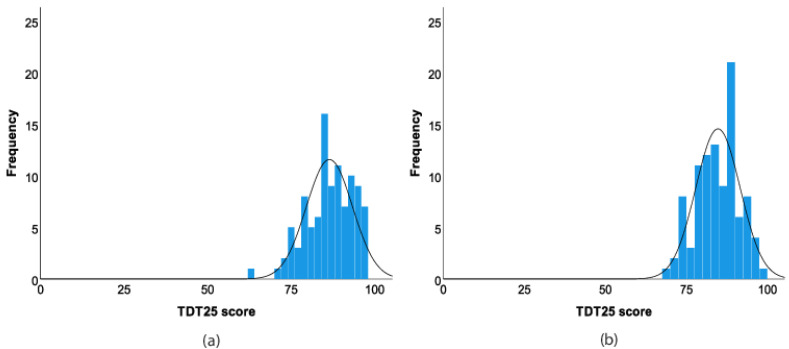
Frequency distributions of Tactile Discrimination Test (TDT) scores for (**a**) dominant and (**b**) non-dominant hands of neurologically healthy participants.

**Figure 3 brainsci-13-00533-f003:**
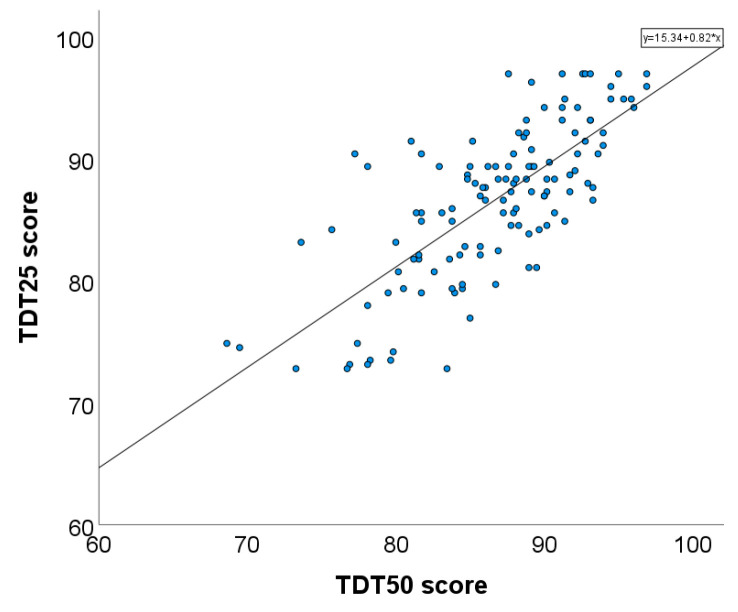
Scatterplot of Tactile Discrimination Test (TDT) scores TDT25 compared to TDT50 (*n* = 63) for the combined dominant and non-dominant hand in neurologically healthy pooled samples.

**Figure 4 brainsci-13-00533-f004:**
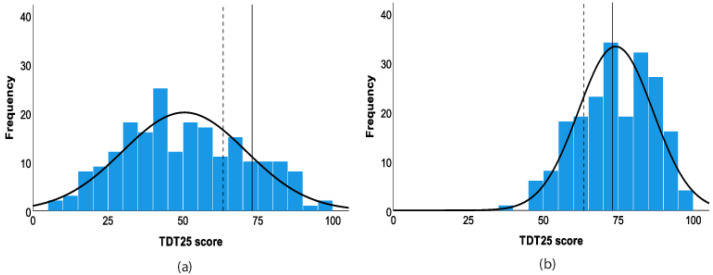
Frequency distributions of tactile discrimination for (**a**) contralesional and (**b**) ipsilesional hands of stroke participants. The solid horizontal lines represent the best estimate for the fifth percentile criterion of abnormality for the TDT25 test version (this criterion was defined from a sample of 100 healthy controls). The dashed lines represent the more conservative criterion of abnormality (zeroth percentile) in which all healthy TDT scores were contained.

**Figure 5 brainsci-13-00533-f005:**
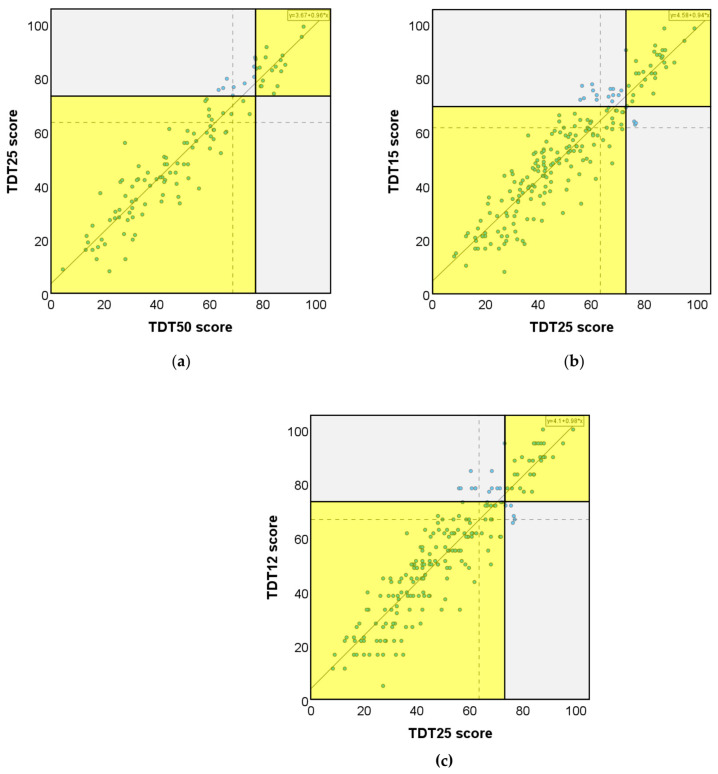
Scatterplot of TDT test scores based on briefer trial presentations when compared to the original test (TDT50) or the full clinical test (TDT25) for the contralesional hand of the pooled sample, i.e., (**a**) TDT25 compared to TDT50, (**b**) TDT15 compared to the TDT25, and (**c**) TDT12 compared to TDT25. The diagonal line is the regression line of no change. The solid vertical and horizontal lines represent the best estimate lines for the fifth percentile criterion of abnormality for each test version (this criterion is defined from the sample of 100 neurologically healthy controls). The dashed line represents the more conservative criterion of abnormality (zeroth percentile) in which all healthy TDT scores were contained for the matched test version. Yellow quadrants represent regions in which scores agree, and grey quadrants are regions where scores show disagreement.

**Figure 6 brainsci-13-00533-f006:**
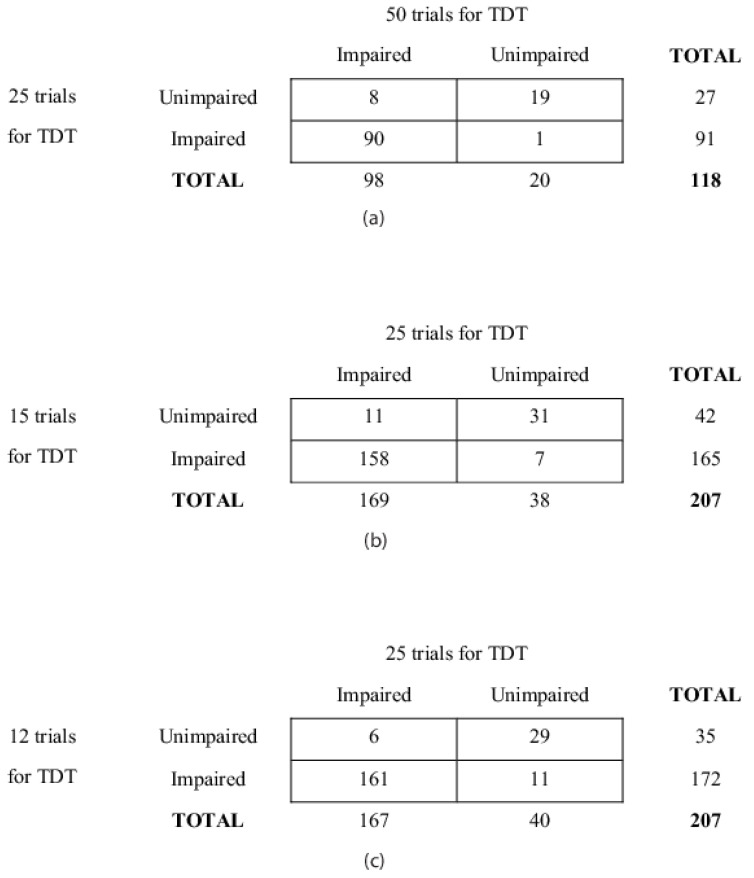
Contingency table analysis of touch discrimination impairment identified in the contralesional hand using the briefer number of trials, i.e., (**a**) TDT25 compared to TDT50, (**b**) TDT15 compared to the TDT25, and (**c**) TDT12 compared to TDT25.

**Table 1 brainsci-13-00533-t001:** Demographic characteristics of stroke survivors from pooled and individual study samples.

Demographic Information	Pooled Sample (*n* = 207)	Discriminative Validity ^a^ (*n* = 50)	SENSe Trial ^b^ (*n* = 46)	IN_TOUCH Trial ^c^ (*n* = 22)	CoNNECT Trial ^d^ (*n* = 45)	NIH Trial ^e^ (*n* = 9)	SENSe CONNECT ^f^ (*n* = 35)
Age, years =							
M (SD)	56.4 (14.5)	52 (14.4)	61.4 (11.9)	59.7 (15.1)	52.8 (14.1)	65.2 (12.6)	56.1 (15.6)
Min, Max	18, 87	18, 79	32, 87	18, 79	26, 82	41, 78	19–81
Gender, *n%*							
Male	144 (70)	36 (72)	33 (72)	14 (64)	32 (71)	6 (67)	23 (66)
Female	63 (30)	14 (28)	13 (28)	8 (36)	13 (29)	3 (33)	12 (34)
Hemisphere affected, *n%*							
Right	95 (46)	21 (42)	20 (43)	8 (36)	20 (45)	5 (56)	21 (60)
Left	109 (53)	29 (58)	26 (57)	14 (64)	23 (51)	4 (44)	13 (37)
Both	3 (1)	0 (0)	0 (0)	0 (0)	2 (4)	0 (0)	1 (3)
Affected side, *n%*							
Dominant	105 (51)	47 (94)	26 (57)	13 (59)	23 (51)	5 (56)	13 (37)
Non dominant	102 (49	3 (6)	9 (41)	9 (41)	22 (49)	4 (44)	22 (63)
Lesion Level, *n%*	(*n* = 198) *						
Cortical	88 (44)	15 (30)	18 (39)	13 (59)	26 (58)	-	16 (29)
Subcortical	55 (28)	9 (18)	13 (28)	8 (36)	15 (33)	-	10 (46)
Both	25 (13)	9 (18)	10 (22)	1 (5)	4 (9)	-	1 (3)
Unknown	30 (15)	17 (34)	5 (11)	0 (0)	0 (0)	-	8 (23)
Stroke type, *n%*							
Ischemic	143 (72)	42 (84)	31 (84)	22 (100)	31 (69)	-	17 (48)
Hemorrhage	52 (26)	8 (16)	15 (33)	0 (0)	14 (31)	-	15 (43)
Unknown	3 (2)	0 (0)	0 (0)	0 (0)	0 (0)		3 (9)

Note: Total number of stroke participants = 207; * Reduced pooled total of *n* = 198, as lesion level was not recorded for the NIH Trial. The frequency and percentage reported in this section refer to the reduced sample size. ^a^ Stroke Discriminative Validity study [23]. ^b^ SENSe = Study of Effectiveness of Neurorehabilitation on Sensation [26]. ^c^ IN_Touch = Imaging Neuroplasticity of Touch [29,30]. ^d^ CoNNECT = Connecting Networks for Everyday Contact Through Touch [27,34]. ^e^ NIH Trial = National Institute of Health [18] stroke participants who underwent additional testing. ^f^ SENSe CONNECT Study [32].

**Table 2 brainsci-13-00533-t002:** Demographic characteristics of neurologically healthy participants from pooled and individual study samples.

Demographic Information	Pooled Sample (*n* = 100)	Normative Validity ^b^ (*n* = 50)	IN_TOUCH Trial ^c^ (*n* = 13)	CoNNECT Trial ^d^ (*n* = 26)	NIH Trial ^e^ (*n* = 11)
Age, years					
M (SD)	52.7 (16.2)	52.1 (12.9)	58.6 (15.3)	49.5 (17.9)	56.1 (22.1)
Min, Max	23, 89	23, 77	23, 79	26, 89	29, 81
Gender, *n*%					
Male	59 (59)	34 (68)	7 (54)	13 (50)	5 (45)
Female	41 (41)	16 (32)	6 (46)	13 (50)	6 (55)
Hand Dominance ^a^					
Right	94 (96)	46 (92)	13 (100)	26 (100)	11 (100)
Left	4 (4)	4 (8)	0 (0)	0 (0)	0 (0)

Note: Total number of neurologically healthy participants = 100; ^a^ Based on Annette questionnaire of hand dominance or Edinburgh Handedness Inventory. ^b^ Normative Validity Study [23]. ^c^ IN_Touch = Imaging Neuroplasticity of Touch [29,30]. ^d^ CoNNECT = Connecting Networks for Everyday Contact Through Touch [27,34]. ^e^ NIH Trial = Toolbox Trial [31] healthy participants who underwent additional testing.

**Table 3 brainsci-13-00533-t003:** Descriptive statistics, criterion of abnormality, and zone of uncertainty from combined dominant and non-dominant hands of the pooled data samples of neurologically healthy older adults for different trial versions of the TDT.

TDT: No. of Trials	Mean PMA	SD	Criterion of Abnormality	
			5th	0th
50 trials ^a^	86.69	5.70	76.98	68.62
25 trials	85.58	6.91	73.10	63.45
15 trials	85.02	8.58	69.45	61.49
12 trials ^b^	89.20	8.45	73.08	66.67

Note: The criterion of abnormality is reported as the fifth percentile. The more conservative criterion under which all TDT scores of healthy controls are contained (zeroth percentile) is also reported. PMA = percent maximum area. SD = standard deviation. SEM = standard error of measurement. LCI and UCI = lower and upper confidence intervals. ^a^ Total number of participants with 50 trial data *n* = 63. Total number of participants with data for the remaining trials is *n* = 100. ^b^ Twelve trial scores are based on responses from four triplet sets instead of the five sets.

**Table 4 brainsci-13-00533-t004:** Descriptive statistics for the contralesional and ipsilesional hand of stroke survivors based on original and brief test versions of the TDT.

No. of Trials	Mean	SD	Minimum	Maximum	Percentiles		
	PMA				25th	50th Median	75th
Contralesional							
50 trials	50.21	22.21	4.48	95.34	30.86	48.62	66.38
25 trials	50.50	20.57	8.28	98.97	34.83	49.31	66.90
15 trials	52.00	20.94	8.05	98.28	37.36	52.30	67.24
12 trials ^a^	53.74	22.06	5.13	100	38.46	53.85	71.80
Ipsilesional							
50 trials	75.71	12.12	29.14	95.52	67.97	76.98	84.78
25 trials	74.14	12.43	35.86	97.93	64.48	74.14	84.14
15 trials	73.81	14.00	28.74	98.28	63.80	75.29	83.91
12 trials ^a^	77.12	14.83	28.20	100	66.67	78.21	89.74

Note: Original test version, TDT50 (*n* = 118); brief test versions, TDT25 trials and under (*n* = 207). SD = standard deviation. ^a^ Twelve trial scores are based on the inclusion of responses from four stimuli sets instead of five sets.

## Data Availability

The data are not publicly available due to planned further analyses.

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
