# Peer review of "Characterizing Touch Discrimination Impairment from Pooled Stroke Samples Using the Tactile Discrimination Test: Updated Criteria for Interpretation and Brief Test Version for Use in Clinical Practice Settings"

_brainsci, 2023, doi:10.3390/brainsci13040533_

Round 1
Reviewer 1 Report
This is an interesting paper with clinical relevance. Pooled data from the Tactile Discrimination Test (TDT) from stroke survivors and neurological healthy adults were re-analyzed in order to characterize touch impairment and to re-establish criterion of abnormality of the TDT. Sensitivity and specificity of a briefer test version of TDT were also determined.
It is overall a well-written paper, and the statistical methods seem sound. I have only a few comments:
Introduction: The Introduction could preferably be shortened somewhat. Some sentences are very long and some information is repeated (for example that the original TDT is an extensive test and takes time to complete).
Material and Method: Study design and samples; please clarify early on that the pooled TDT data comprised data from both stroke survivors and from neurological healthy people.
Results: Table 2, most participants in the different studies were men. However, in the pooled sample column only 29% were mean. Please check the figure. Furthermore, on page 10 (Associations between test scores for the different TDT versions), it is stated that not all participants in the various studies completed the TDT50. Thus, clarify how many participants that were included in this analysis.
Author Response
Please see the attachment -response letter and revised manuscript with track changes.

Reviewer 2 Report
In the presented manuscript the authors made an attempt to explore and standardize method of tactile discrimination testing (TDT) following stroke events. By pooling data from different studies they extracted baseline data from 207 stroke survivors and 100 older neurologically healthy controls. Reanalyzed data compared TDT Percent Maximum Area (PMA) for sensitivity and specificity between different versions of TDT (25 vs 50 trials; 12 or 15 vs 25 trials). Touch sensitivity was impaired in contralateral to lesion hand in 83% of patients, while it was also impaired in ipsilateral to the lesion hand in 42% of cases, suggesting that use of ipsilateral to lesion hand cannot be reliable standard for testing. Based on the analysis the authors concluded that data support the use of shorter version of TDT in clinical setting.
Author Response
Thank you for the positive feedback.
Reviewer 3 Report
In summary, it would be advisable to specify each subsection for better reading.
The introduction is written in a very long way, it should be cut and restructured.
The conclusions must be more synthesized. The rest of the manuscript has been well done by the authors and could advance this pathology.
Author Response
Please see the attachment with response to reviewers and revised manuscript with track changes.
